# Localization of Chicken Rab22a in Cells and Its Relationship to BF or Ii Molecules and Genes

**DOI:** 10.3390/ani13030387

**Published:** 2023-01-23

**Authors:** Fengmei Yu, Muhammad Akmal Raheem, Yang Tan, Muhammad Ajwad Rahim, Lisha Zha, Jun Zhang, Zhiwei Zhu, Zhonghua Li, Fangfang Chen

**Affiliations:** 1Key Laboratory of Veterinary Pathobiology and Disease Control, College of Animal Science and Technology, Anhui Agricultural University, Hefei 230036, China; 2Institute of Biopharmaceutical and Health Engineering, Tsinghua Shenzhen International Graduate School, Tsinghua University, Shenzhen 518055, China; 3Tsinghua-Berkeley Shenzhen Institute, Tsinghua Shenzhen International Graduate School, Tsinghua University, Shenzhen 518055, China; 4Department of Zoology, Ghazi University Dera Ghazi Khan, Punjab 32200, Pakistan

**Keywords:** cRab22a, BFa, cIi, co-localization, expression

## Abstract

**Simple Summary:**

Rab22a is an important molecule in endocytosomal transport, while BF and cIi areantigen-presenting presenting molecules in poultry. The relationship between Chicken Rab22a (cRab22a) and BF/cIi is unclear. In this study, the intracellular localization of cRab22a and its key domains and amino acids were determined by gene mutation and immunofluorescence methods, then the relationship between cRab22a and BF/cIi genes and molecules was studied by RT-PCR and Co-IP methods. The results showed that the Rab22a amino acid sequences of different species are highly conserved. Positions Ser41 and Tyr74 in Switch region are key amino acids for intracellular localization. Both low expression and high expression of cRab22a genes will affect the expression of BF gene, while only low expression will affect the expression of cIi gene. The cRab22a molecule does not directly interact with the BF/cIi molecules. This study clarified the relationship between cRab22a and BF or cIi, and provided an experimental basis for further research on the intracellular transport of BF or cIi and related immune response.

**Abstract:**

Rab22a is an important small GTPase protein the molecule that is involved in intracellular transportation and regulation of proteins. It also plays an important role in antigens uptake, transportation, regulation of endosome morphology, and also regulates the transport of antigens to MHC (Major Histocompatibility Complex) molecules. To investigate the role of Rab22a, the intracellular co-localization of chicken Rab22a (cRab22a) molecule and its relationship to BF and chicken invariant chain (cIi) molecules was studied. A 3D protein structure of Rab22a was constructed by using informatics tools (DNASTAR 4.0 and DNAMAN). Based on the model, the corresponding recombinant eukaryotic plasmids were constructed by point mutations in the protein’s structural domains. HEK 293T cells were co-transfected with plasmids pEGFP-C1-cIi to observe the intracellular co-localization. Secondly, the DC2.4 Mouse Dendritic Cell and Murine RAW 264.7 cells were transfected with recombinant plasmids of pmCherry-cRab22a and pmCherry-mRab22a respectively. Subsequently, the intracellular localization of cRab22a in early and late endosomes was observed with specific antibodies against EEA1 and LAMP1 respectively. For gene expression-based studies, the cRab22a gene was down-regulated and up-regulated in HD11 cells, following the detection of transcription levels of the BFa (MHCIa) and cIi genes by real-time quantitative PCR (RT-qPCR). The interactions of the cRab22a gene with BFa and cIi were detected by co-immunoprecipitation (Co-IP) and Western blot. The results showed that the protein structures of chicken and mouse Rab22a were highly homologous (95.4%), and both localize to the early and late endosomes. Ser41 and Tyr74 are key amino acids in the Switch regions of Rab22a which maintain its intracellular localization. The down-regulation of cRab22a gene expression significantly reduced (*p* < 0.01) the transcription of BFa (MHCIa) and cIi in HD11 cells. However, when the expression of the cRab22a gene was increased 55 times as compared to control cells, the expression of the BFa (MHCIa) gene was increased 1.7 times compared to the control cells (*p* < 0.01), while the expression of the cIi gene did not significantly differ from control (*p* > 0.05). Western blot results showed that cRab22a could not directly bind to BFa and cIi. So, cRab22a can regulate BFa and cIi protein molecules indirectly. It is concluded that cRab22a was localized with cIi in the endosome. The Switch regions of cRab22a are the key domains that affect intracellular localization and colocalization of the cIi molecule.

## 1. Introduction

The major histocompatibility complex (MHC) classes I and II are important antigen-presenting molecules (present antigens to T cells) to initiate immune responses. Different species have different names for MHC molecules, the chicken MHC is known as BF molecules which are structurally and functionally similar to other species. It was confirmed by immunoprecipitation and gel electrophoresis of radio labeled molecules that, the BF and BL antigens are similar to class I and class II molecules respectively [1]. During the process of antigen-presenting, the Ii chaperone of MHC class II molecules mainly assists in the processing and presentation of antigenic peptides of class II molecules [2,3]. The Ii molecules are also involved in the cross-presentation of antigens by MHC class I molecules [4]. The endosomes of immune cells are important organelles in the processing of antigens by MHC/Ii molecules. The MHC molecules, Ii molecules, and the endosome correlate with each other. The Ii molecules are expressed when endosomes are enlarged and their maturation and protease degradation are delayed. This facilitates the processing of antigens and their binding to MHC molecules [5]. At the same time, Ii molecules are also responsible for the categorization of MHC class I and II molecules into the corresponding endosomal pathways and for regulating B-cell development and maturation [6,7].

The localization of MHC and its chaperone molecules to the endosome requires the participation of a variety of regulatory molecules, including several small GTPases [8]. These small membrane-associated GTPases (Rabs) are molecular switches that control biochemical pathways and regulate active (GTP binding) and inactive (GDP binding) states through an “on/off” mode [9]. These small Rabs proteins are a class of important pivot molecules in this family that specialize in intracellular protein transportation [10]. The Rabs proteins recruit specific effectors for the transportation and organization of various organize membrane micro-domains and alter their multiple membrane transport steps [11]. Endosomes and lysosomes are important organelles that contain lysosomal proteases enzymes, that play important role in antigen-processing and antigen-presenting molecules [12,13].

A variety of Rabs, such as Rab22a, Rab7b, and Rab5a are involved in the transport of antigens and antigen-presenting molecules through the endosomes of antigen-presenting cells (APC) [14]. Rab22a is a subfamily member of Rab5, and the amino acid sequence homology is up to 52%. [15]. The Rab22a mainly exists in early endosomes (E/SEs) and recycling endosomes (REs) which affects clathrins and nonclathrins endocytosis pathway to regulate the transport of viral antigens [16,17,18,19,20].

Rab22a is an important regulatory protein that promotes the assembly of the BLOC1-BLOC2-KIF13A complex on early (sorting) endosomes (E/SEs) to generate recycling endosomes (REs) that maintain homeostasis of cells and their organelles. It also regulates the morphology of endosomes [16,21]. Rab22a plays an important role in antigen cross-presentation by regulating the transport of different types of antigens on antigen-presenting cells by acting on the antigen-presenting molecules MHC I and CD1a [22,23]. The Rab22a stabilizes the intracellular pool of MHC I molecules, and transfers these molecules to phagosomes for their restoration to the cell surface [23]. It can also regulate the transport of transferrin (iron) receptors, the uptake of antigens (such as viruses and spirochetes), and the formation of T-cell complexes to affect antigen presentation [18,19,24]. As the principal regulators of intracellular membrane trafficking, Rab proteins are required at the different steps of antigen processing, presentation, and loading onto MHC class I molecules, including in the maturation of early endosomes (Rab5a) and late endosomes (Rab7) [25]. Rab7b is required for normal lysosome function and in particular, it is an essential factor for the retrograde transport of molecules from endosomes to the trans-Golgi network (TGN) [26]. Rab5a is also involved in the regulation of antigens and antigen-presenting molecules, for example, simvastatin (an inhibitor of Rab5a) preferentially affected the invariant chain-dependent MHC class II pathway [27].

The MHC and Ii molecules are the key molecules that are involved in antigen presentation, and endocytosomes and lysosomes are important organelles for MHC and Ii molecules and antigen presentation. Rab22a is also an important molecule associated with these molecules. Therefore, we selected Rab22a (*cRab22a*) and chicken MHC I (*BF*), and Ii (*cIi*) molecules in poultry to investigate their structure, characterization, and intracellular localization to clarify the functional binding domains of these associated molecules and their relationships with antigen-presenting molecules for antigen cross-presentation.

## 2. Materials and Methods

### 2.1. Source of Cells, Plasmids, and Other Materials

All plasmids pCMY-Myc, pmCherry-C1, and PEGFP-C1 were purchased from Clontech Company (Dalian, China). DC2.4 Mouse Dendritic Cell and Murine RAW 264.7 cells, chicken macrophage cells HD11, and human embryonic kidney cells (HEK 293T) were purchased from the American type culture collection (ATCC). Australian Fetal Bovine Serum (FBS) and DMEM medium were purchased from Hyclone Biotechnology Co., Ltd. (Logan, Utah, USA); mouse anti-EEA1 antibody, mouse anti-LAMP1 antibody, and fluorescein-labeled rabbit anti-mouse IgG secondary antibody was purchased from CST Biotechnology Co., Ltd. (USA).

### 2.2. Construction of Recombinant Plasmids

All primers were designed from the reference sequence of chicken Rab22a (cRab22a) (ID: NM_001030867) and mouse Rab22a (mRab22a) (ID: NM_024436) gene with the help of Oligo 6.0-based in the GenBank database. Using cDNAs templates of laboratory-maintained chicken and mouse, the eukaryotic recombinant plasmids pmCherry-C1-*cRab22a*, pmCherry-C1-*mRab22a*, and pCMV-Myc-*cRab22a* were constructed with PCR enzyme (Takara, Dalian, China) and restriction endonuclease (Takara, Dalian, China) (Table 1). Using pmCherry-C1-*cRab22a* plasmid as a template of the eukaryotic recombinant plasmid of mutant cRab22a domains that were constructed with overlapping extension PCR and continuous cloning. The mutant functional regions targeted (or the amino acids) and the primers used are shown in Table 1. Upstream (F) and downstream primers (R1) for *cRab22a* were used to amplify the first segment. Again upstream and downstream primers R and F1 for *cRab22a* were used to amplify the second segment. The amplified fragment was recovered by agarose gel electrophoresis.

Using the two recovered PCR products as templates, the primers F and R of cRab22a were also used to amplify the *cRab22a* mutants. We used the Seamless Cloning system (Aidlab, Beijing, China) to clone the PCR products in a 10 μL reaction volume containing 5 μL of 2× OneStep Cloning Mix, 3 μL of double-digested, and 2 μL of double digested empty vector. The reaction of the cells has done at 50 °C for 30 min and then transformed into *Escherichia coli* DH5α competent cells. The recombinant bacteria were initially identified by PCR and the plasmids were then sequenced by Tsingke (Nanjing, China).

### 2.3. Alignment of Chicken and Mouse Amino Acid Sequences and 3D Protein Structures

DNASTAR 4.0 and DNAMAN software were used to analyze the homology between the chicken and mice Rab22a proteins. The protein sequences were submitted to Swiss-Model (https://swissmodel.expasy.org/interactive) for homology modeling, and the 3D structure of the Rab22a protein was constructed with the PyMOL1.8 software.

### 2.4. Gene Overexpression and Silencing

Chicken macrophage cells HD11 were transfected by the recombinant plasmid pmCherry-C1-*cRab22a* and the empty plasmid pmCherry-C1 (used as control) with Xfect^TM^ Transfection Reagent (Takara, Dalian, China), and cultured for 48 h to obtain gene overexpression. The *cRab22a* gene was silenced by the transferring of small interfering RNA (siRNA-cRab22a) (sequence: 5′-GTCAGCAGCTGCCATTATA-3′) (RiboBio, Guangzhou, China). The siRNA-cRab22a was inserted into HD11 cells and incubated for 36–48 h. The total RNA from the *cRab22a*-overexpressed and the *cRab22a*-silenced cells was extracted and reverse-transcribed. The transcription levels of *cRab22a*, *BFα*, and *cIi* genes in the chicken cells were determined with real-time quantitative PCR (RT-qPCR). The primer sequences used to amplify *cRab22a* were F: 5′-CAGCTGGACAGGAACGGTTT-3′ and R: 5′-ACAATGTTTGGAGGTCCGTGT-3′. The primers for the chicken *BFα* and *cIi*, and the internal reference gene of glyceraldehyde phosphate dehydrogenase (*GAPDH*) were selected from previous studies [28].

### 2.5. Intracellular Localization

The day before transfection, DC2.4 and RAW264.7 cells were cultured in a 24-well plate with a density of 1 × 105/well in a complete culture medium containing Dulbecco’s modified Eagle’s medium (DMEM) (Biosharp, Beijing, China) 10% fetal bovine serum (FBS) (Hyclone Laboratories, Inc., Logan, UT, USA) and 1% antibiotic (mixture of penicillin and streptomycin) (Biosharp) for overnight at 37 °C under 5% CO_2_. The DC2.4 and RAW264.7 cells were transfected with eukaryotic recombinant plasmids such as pmCherry-C1-cRab22a, pmCherry-C1-mRab22a, and pmCherry-C1 (control) respectively. The cells were washed with Phosphate-buffered saline (1XPBS) after 24–48 h of transfection. Then these transfected cells were dipped into 4% paraformaldehyde for 10 min (Saibo, Beijing, China). After that, the cells were again washed with 1XPBS following the incubation at room temperature for 1 h by adding 0.3% Triton X-100 with PBS (Beyotime, Shanghai, China) and 5% goat serum (Beyotime). Mouse anti-EEA1 and anti-LAMP1 antibodies (CST, MA, USA) were added to the cultures at the ratio of 1:100 and then incubate at 4 °C overnight. The unbounded antibodies were removed by washing the cells with 1XPBS. The fluorescein isothiocyanate labeled (FITC-labeled) goat anti-mouse IgG secondary antibodies (CST) was added to the cultures at the ratio of 1:1000 and then incubated at room temperature for 2 h. Again cells were washed with 1XPBS, then added 4′,6-diamidino-2-phenylindole (DAPI) (Solarbio, Beijing, China) in the cell culture and leave it for 10 min to stain their nuclei. DAPI staining was done to visualize nuclear DNA in both living and fixed cells. DAPI staining was used to determine the number of nuclei and to assess gross cell morphology. After staining, cells were washed three times with 1XPBS. Following light microscopic analyses, the stained cells were processed for mounting, and observation under a laser confocal microscope.

### 2.6. Confocal Observation

The HEK 293T cells were transfected with mutant eukaryotic plasmid pmCherry-C1-*cRab22a* (Express Red-fluorescence-conjugated protein) and the recombinant plasmid PEGFP-C1-*cIi* (Express Green-fluorescence-conjugated protein). After 24–48 h of culture, the cells were washed three times with 1XPBS following fixation with 4% paraformaldehyde for 10 min. After three times washing with 1XPBS, and then stained with DAPI for 10 min, again washed three times with PBS, and finally sealed. The localization of cRab22a and mutant cRab22a with cIi in the cells was observed with laser confocal microscopy.

### 2.7. Co-Immunoprecipitation (Co-IP) and Western Blotting

The constructed plasmid pCMV-Myc-*cRab22a*, pCMV-Myc, pEGFP-C1-*cIi*, pEGFP-C1-*BFα*, and pEGFP-C1 were transfected to the HEK 293T cells following incubation for 48 hrs and washed three times with 1XPBS. After that, 200 μL of Co-IP cell lysis solution containing phenylmethylsulfonyl fluoride (PMSF) (Biolinkedin, Shanghai, China) was added to each well for cell lysis. The mixture obtained after cell lysis was placed on ice for 20 min, then centrifuged at 12,000× *g* for 10 min, and the supernatant was collected. These sediments were used as a positive control and treated with anti-Myc-antibody-conjugated magnetic beads (Biolinkedin) following incubation for 12 h and the supernatant was discarded carefully. The remaining magnetic beads were washed with pre-cooled cell lysis solution and added 40 μL of loading buffer. The protein samples were boiled at 100 °C for 10 min and then treated with labeled antibodies (anti-Myc and anti-green fluorescent protein [GFP]) (Abmart, Shanghai, China) for western blotting to detect protein-protein interactions.

### 2.8. Statistical Analysis

The statistical analysis was performed by SPSS 16.0 (SPSS Inc., Chicago, IL, USA). The results of all experiments were analyzed by *t*-test, followed by Tukey’s HSD (honestly significant differences) for post-hoc testing to compare the significance (P) between the means of different groups. The differences were considered statistically significant at *p* < 0.01.

## 3. Results

### 3.1. Amino Acid Sequences Alignment of cRab22a Protein and mRab22a Protein and Construction of Their 3D Structures

The analyses showed that the Rab22a proteins of both species contain 194 amino acids and were highly homologous (95.4%). However, only nine amino acids were different (labeled with red color) located at positions 101, 110, 123, 131, 171, 174, 176, 188, and 189 (Figure 1). The amino acids at key sites are identical to the GTP binding site (marked with ★), and the key domains are highly consistent such as the Switch I region (greenish background) and the Switch II region (wavy underlining) (Figure 1).

The amino acid sequences of cRab22a and mRab22a were further analyzed by SWISS-MODEL to construct 3D structures. It was found that the 3D structures of both proteins were similar, containing 6 β-folds and 5 α-helices. These results indicate that cRab22a and mRab22a are similar in protein structure with main structural domains and key amino acids (Figure 2).

### 3.2. Co-Localization of cRab22a in Early and Late Endosomes

The endosomes of immune cells are closely related to the process of antigen presentation by MHC molecules, while cRab22a is related to the migration of MHC molecules. We used immunofluorescence and a laser confocal technique to observe and understand the process of localization (location) of cRab22a in early and late endosomes. The early endosome marker (EEA1) and late endosome/lysosomal marker (LAMP1) were stained with green-specific fluorescence-labeled antibodies, and target proteins (cRab22a and mRab22a) were labeled with a red fluorescent protein (RFP) (RFP/cRab22a and RFP/mRab22a), and the nuclei of cells were stained with blue dye (DAPI) (Figure 3). When the labeled proteins co-localized intracellular organelles, the merging of green (endosome) and red (cRab22a and mRab22a) colors appeared in orange color showed signals of orange color (Figure 3, shown by arrow).

The cRab22a and mRab22a co-localized with both early and late endosomes (EEA1 and LAMP1) respectively, so the merging of Rab22a (red color) with endosomes (green color) appeared in orange color signals (Figure 3A,D; Figure 3B,E). The RFP empty vector was used as a negative control, it did not co-localize with endosomes and that’s why no orange signals were detected. These results indicate that cRab22a and mRab22a were localized in early and late endosomes (Figure 3C,F).

### 3.3. Intracellular Co-Localization of Recombinant Plasmids of cRab22a and Its Mutants with cIi

The structural key domains and amino acids of cRab22a were mutated in Switch I and Switch II to observe changes in intracellular co-localization of these mutants with cIi molecules. Key domains and amino acids influence the role of cRab22a in intracellular localization. It was observed that the key domains and specific amino acids have a great influence on the intracellular localization of cRab22a. The plasmids encoding mRab22a, cRab22a, and its mutants such as cRab22a_(M36–41aa)_, cRab22a_(M42–46aa)_, cRab22a_(M70–75aa)_, cRab22a_(M74–75aa)_, cRab22a_(Q64L)_, cRab22a_(Y75A)_, cRab22a_(Y74A)_, and cRab22a_(S41A)_ were generated by PCR. Agarose gel electrophoresis showed that all target fragments were 585 bp, consistent with the expected size. Then these fragments were inserted into the eukaryotic plasmid pmCherry-C1 to construct 10 relevant recombinant eukaryotic plasmids. The DNA sequencing confirmed that all these plasmids contained the corresponding target gene sequences.

Finally, HEK 293T cells were co-transfected with relevant recombinant eukaryotic plasmids and their colocalization was observed. Switch I and II are important domains of cRab22a. The wild-type cRab22a was mainly located (localized) outside of the nucleus (Figure 4A). The intracellular localization of the mutant RFP/cRab22a_(M36–41aa)_ was done in domain Switch I by the replacement of amino acids from the 36–41 position, and alteration was distributed throughout the cell but did not co-localize intracellular with GFP/cIi (no orange granules) (Figure 4B). The amino acids 42–46 were replaced by mutant RFP/cRab22a_(M42–46aa)_, and this mutant was located in the cytoplasmic region and also co-localization (combine) with GFP/cIi (observed orange granules) (Figure 4C). The intracellular distribution of the mutants RFP/cRab22a_(M70–75aa)_ and RFP/cRab22a_(M74–75aa)_, was done in the Switch II domain in which amino acids 70–75 and 74–75 were replaced, but they did not co-localize with GFP/cIi in the cells (no orange granules) (Figure 4D,E).

To determine the key amino acids of cRab22a that are essential for its localization, the above-mentioned mutants were further investigated. The results showed that both RFP/cRab22a_(Q64L)_ and RFP/cRab22a _(Y75A)_ co-localized with GFP/cIi in the cytoplasm which was identified by orange a yellow color (Figure 4F,G). However, the intracellular distribution of the mutants RFP/cRab22a_(S41A)_ and RFP/cRab22a_(Y74A)_, was done by the replacement of single amino acids 41 and 74, but they did not co-localize with GFP/cIi in the cells (no orange granules) (Figure 4H,I). These results indicate that Switch I and II are the key domains of Rab22a, while the amino acids at the position of 41 and 74 are also essential amino acids for maintaining their active structure.

### 3.4. Effects of cRab22a Gene Silencing and Overexpression of BFα and cIi Gene

The BF and cIi function synergistically by affecting each other at the gene level. However, it was not clear whether either expression of BF and cIi correlates or not with the *cRab22a* gene. Therefore, siRNA-cRab22a and pmCherry-C1-*cRab22a* were used to silence *cRab22a* expression and overexpress their respective gene. The RT-qPCR was used to determine the transcription levels of *BF* and *cIi* in the same cells.

After the transfection of pmCherry-C1-cRab22a into HD11 cells, the transcription level of cRab22a was increased 55 times as compared to the control group (*p* < 0.01). This increase in cRab22a transcription level up-regulates the *BFα* gene by 1.7 times as compared to the control group (*p* < 0.01). Whereas the expression level of the *cIi* gene was not significant (*p* > 0.05) (Figure 5A). The results showed that the specific siRNA significantly down-regulates the transcription level of *cRab22a* in HD11 cells, which was only 35.0% in the control group (*p* < 0.01). Whereas the transcription level of the *BFα* and *cIi* genes decreased up to 53.7% and 37.0% respectively as compared to the control group (*p* < 0.01) (Figure 5B).

These findings suggest that down-regulation and overexpression of *cRab22a* affect the transcription level of *BFα*. The down-regulation of cRab22a reduces the expression of *cIi*, while the overexpression of *cRab22a* did not affect the transcription level of *cIi*.

### 3.5. Interaction of cRab22a with BF and cIi Molecules

The cRab22a interacts with the BF and cIi molecules at the genetic level, but the lack of background enforces us to use the Co-IP method to determine the relationships of cRab22a with BFα and cIi molecules. The Myc/cRab22a was mixed with GFP/BFα, GFP/cIi, and GFP (control), following the addition of Myc-directed magnetic beads. The mixture was incubated to obtain cognate molecules. To identify the linkage of these molecules, complexes were washed and stained with specific antibodies. The western blotting technique identified that the Myc/cRab22a molecule remained separate and it did not make any complex with GFP/BFα (Figure 6A) and GFP/cIi (Figure 6B). In order to verify the presence of GFP/BFα, GFP/cIi, and GFP in the reaction samples, the unwashed sample was used as a control for the detection of antibodies after incubation. Our findings showed that only GFP/BFα (Figure 6A) and GFP/cIi molecules (Figure 6B) appeared in specific bands up to their corresponding point. Therefore, Myc/cRab22a did not make any complex with GFP/BFα and GFP/cIi molecules.

## 4. Discussion

The molecular structure of Rab22a protein in different species is highly conserved but their main properties are similar. In this study, we found that the homology between the amino acid sequence of cRab22a and mRab22a was 95.4%, and even their 3D structural and GTP binding site of the Switch I and II domains were also similar (Figure 1 and Figure 2). Moreover, the early and late endosome localization properties of Rab22a were consistent in both animals (Figure 4).

The previous studies have shown that the Switch structure is a key domain that determines and maintains the stability of Rab22a molecule, in which Phe42 and Trp59 are the key amino acids of the Switch I domain and Leu70 and Met73 are the key amino acids of Switch II domain that determine the hydrophobicity of Rab22a protein [29]. Through this study, we came to know that the Switch I and Switch II regions are the key domains of the Rab22a molecule for its intracellular localization through point mutation. Ser41 and Tyr74 are the key amino acids of Switch I and Switch II domains respectively, and the mutation of these amino acids altered the intracellular localization characteristics of that protein (Rab22a) (Figure 4). In this study, we confirmed that cRab22a with cIi co-localize in the endosome, and the key amino acids in Switch I domains (Ser41) and Switch II domains (Tyr74) can affect its co-localization process (Figure 3). Chicken MHC class I molecules also known as BF molecules are composed of α and β chains, and these chains are important for the determination of BF molecules’ functions (antigen presentation) and characteristics (polypeptide) [30,31,32]. In our previous studies, it was found that BFa can be co-localized with Ii in the endocytosomes, so we believe that cRab22a can also co-locate with BFa in the endocytosomes [33], and the amino acids that affect the localization of cRab22a in cells will also affect its co-localization with BFa in the endocytosomes.

It was also found that BFα and cIi were influenced by cRab22a at the gene level. The change in the transcription level of cRab22a significantly reduced the transcription of cIi and BFα in HD11 cells (*p* < 0.01) (Figure 5A), whereas the up-regulation of cRab22a expression also significantly affect the transcription of BFα but not cIi (Figure 5B). Interference with the expression of Rab22a may impair the transport and expression of MHCI, as has been found Rab22a regulates the recycling of membrane proteins internalized independently of clathrin [23]. In this study, we confirmed that cRab22a with BF and cIi co-localize in the endosome, and the key amino acids in Switch I domains (Ser41) and Switch II domains (Tyr74) can affect its co-localization process (Figure 3). It was also found that *BFα* and *cIi* were also influenced by *cRab22a* at the gene level. The change in the transcription level of *cRab22a* significantly reduced the transcription of *cIi* and *BFα* in HD11 cells (*p* < 0.01) (Figure 5A), whereas the up-regulation of *cRab22a* expression also significantly affect the transcription of *BFα* but not *cIi* (Figure 5B). Interference with the expression of Rab22a may impair the transport and expression of MHCI, as has been found Rab22a regulates the recycling of membrane proteins internalized independently of clathrin.

The overexpression of *the cRab22a* gene has no significant effect on the transcription level of *cIi* due to the lack of transportation of proteins in the cell. Further Co-IP experimental results confirmed that the cRab22a did not affect indirectly on *cIi*. Therefore, the interactions of cRab22a with *BFα* and *cIi* molecules are regulated indirectly by some intermediate proteins. The regulatory relationship between cRab22a and cIi may affect endosomal trafficking, cell migration, and cellular signaling. This would benefit from some literature review of potential signaling pathways involved that may be affected by endosome pathway disruption, e.g., the transcription factors downstream of invariant chain-MIF. Because the MIF/invariant chain axis has been found to stimulate rapid and sustained ERK1/2 signaling, PI3K/AKT signaling, cell proliferation, and anti-apoptotic functions of MIF [34,35,36,37,38].

## 5. Conclusions

The results of this study show that the structure of Rab22a is highly conserved among species and was localized in both early and late endosomes. The Ser41 and Tyr74 are the key amino acids in the Switch I and Switch II regions that determine the endosomal localization of Rab22a. The down-regulation of c*Rab22a* significantly affects the transcriptional level of the *cIi* and *BFα* genes, while the up-regulation of *cRab22a* significantly affects the transcriptional level of the *BFα* gene, not cli. cRab22a does not bind to BFα and cIi molecules. The cRab22a can also co-localize with cIi molecules in the endosomes.


**Main Findings**
Intracellular localization of chicken Rab22a (cRab22a) in early and late endosomes can be observed with specific antibodies against EEA1 and LAMP1 respectively. Meanwhile, cRab22a can be co-localized with cIi in the endocytosisThe protein structures of chicken and mouse Rab22a were highly homologous (95.4%), and both localize to the early and late endosomes. Ser41 and Tyr74 are key amino acids in the Switch regions of Rab22a which maintain its intracellular localization.Interference with the expression of the cRab22a gene can down-regulate the expressions of *BFa* and *Ii* genes, while high expression of the cRab22a gene can affect the expression of *BFa* gene but not Ii. As well, cRab22a cannot interact directly with these two molecules.


## Figures and Tables

**Figure 1 animals-13-00387-f001:**
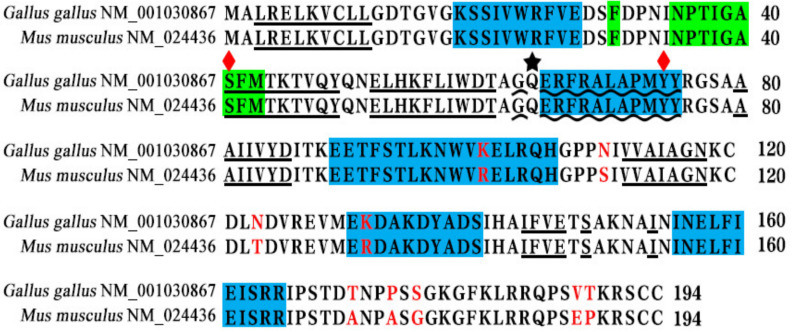
Aligned amino acid sequences of cRab22a and mRab22a indicated that Red-labeled represents mutated ones; underlining indicates β-folds; blue background indicates α-helices; green background indicates the Switch I area; wavy underlining indicates the Switch II area; black star ★ indicates GTP-binding site; red diamonds indicate the point mutations.

**Figure 2 animals-13-00387-f002:**
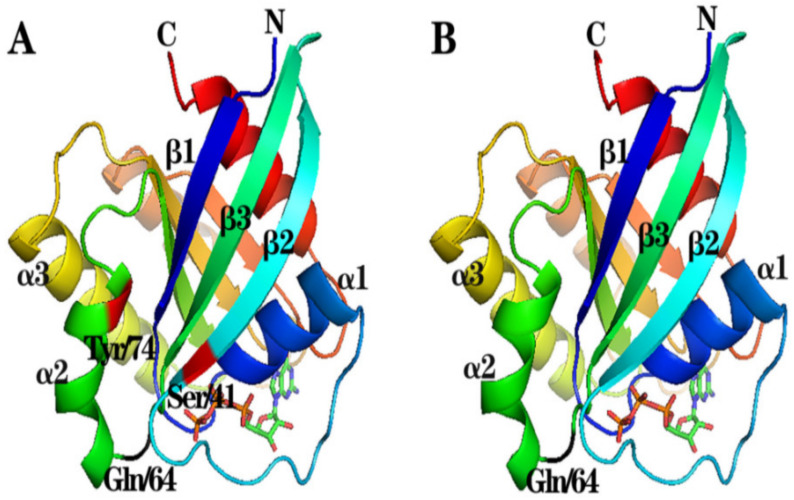
(**A**) 3D protein structures of cRab22a and (**B**) 3D protein structures of mRab22a. Red-labeled Ser41 and Tyr74 are the mutated amino acid sites, black labeled Gln64 is the GTP-binding site.

**Figure 3 animals-13-00387-f003:**
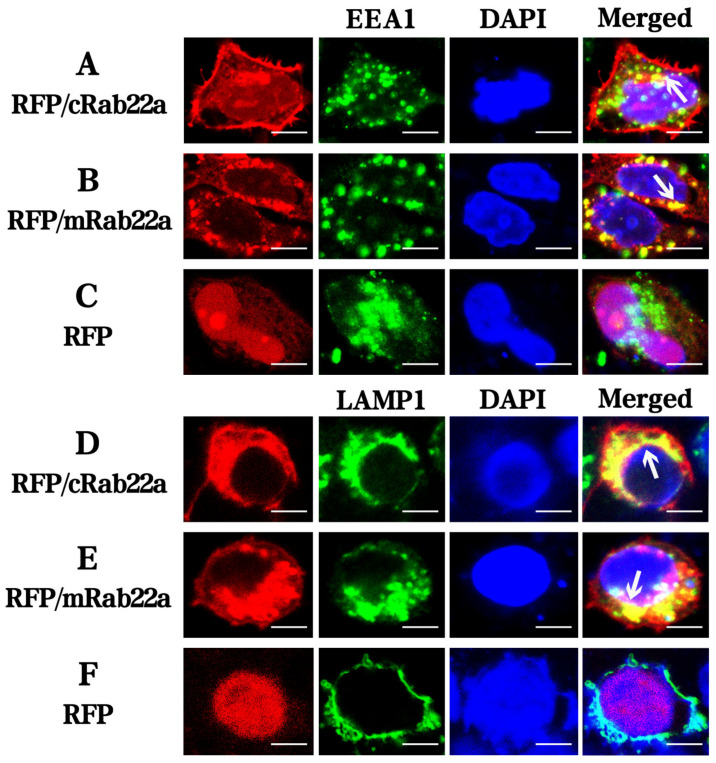
Co-localization of RFP/cRab22a and RFP/mRab22a with EEA1 and LAMP1 in eukaryotic cells (400×). The arrows indicate the co-localization of molecules (50 μm).

**Figure 4 animals-13-00387-f004:**
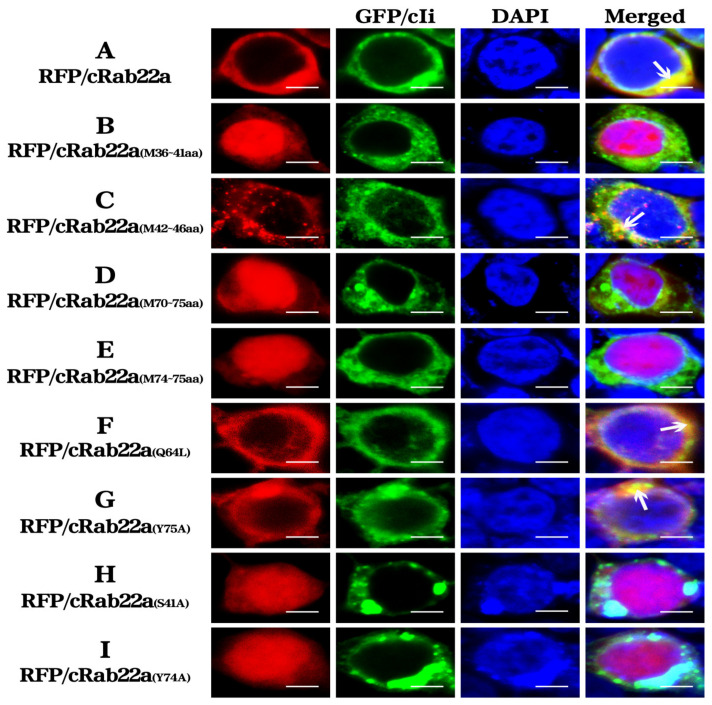
Co-localizations of RFP/cRab22a and its mutants with GFP/cIi in eukaryotic cells (400×). The arrows indicate the co-localization (linkage) of molecules (50 μm).

**Figure 5 animals-13-00387-f005:**
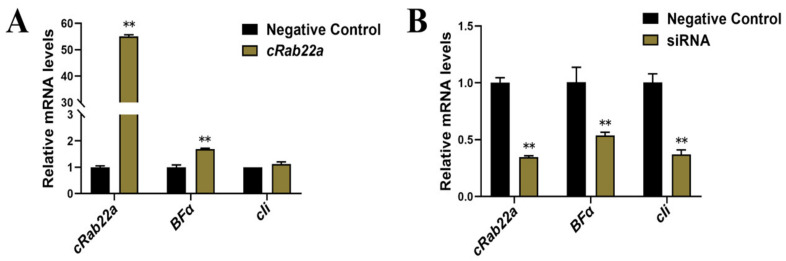
(**A**) The effects of *cRab22a* overexpression, and (**B**) the effect of *cRab22a* silencing (down-regulation) on the transcription level of *BFα* and *cIi*, ** indicates a highly significant difference (*p* < 0.01).

**Figure 6 animals-13-00387-f006:**
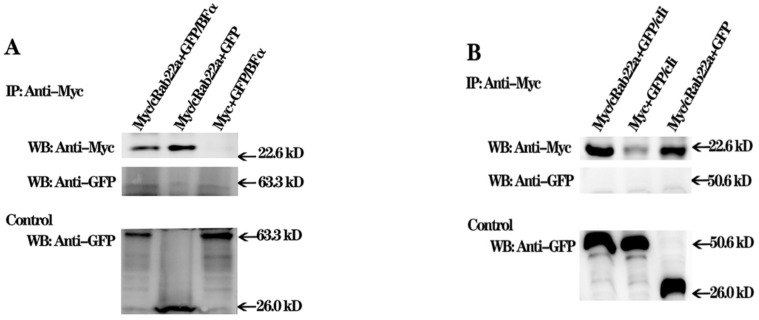
Western blotting for the detection of linkage or binding of Myc/cRab22a with GFP/BFα and GFP/cIi. (**A**)**;** The detection of linkage or binding of Myc/cRab22a with GFP/BFα. (**B**)**;** The detection of link-age or binding of Myc/cRab22a with GFP/cIi.

**Table 1 animals-13-00387-t001:** Sequences of primers used in this study.

Amplification Region or Substituted Amino Acids	Functional Region	Primer Sequences (5′-3′)	Enzymes	Recombinant Plasmids (585bp)	Templates
cRab22a ORF		F: GTCCGGACTCAGATCTCGAGCTATGGCTCTGAGGGAGCTGAAA R: TATCTAGATCCGGTGGATCCTCAACAGCAGCTGCGCTTTGT	Xho I Bam H I	pmCherry-C1-cRab22a	cDNA
mRab22a ORF		F: ACGCCTCGAGCTATGGCGCTGAGGGAAC R: AGAGAGTCGACTCAGCAGCAGCTTCGC	Xho I Sal I	pmCherry-C1-mRab22a	cDNA
41	Switch I region	F1:CAATAGGAGCCGCCTTTATGACCAAG R1:TGGTCATAAAGGCGGCTCCTATTGTT		pmCherry-C1-cRab22a_(S41A)_	pmCherry-C1-cRab22a
74	Switch II region	F1: TTTAGCTCCAATGGCGTATAGAGGGTCA R1:GACCCTCTATACGCCATTGGAGCTAAAG		pmCherry-C1-cRab22a_(Y74A)_	pmCherry-C1-cRab22a
64	β-folds	F1:ACAGCTGGATTAGAACGGTT R1:AACCGTTCTAATCCAGCTGT		pmCherry-C1-cRab22a_(Q64L)_	pmCherry-C1-cRab22a
36–41	Switch I region	F1:CAACATCAACGCCGCAGCTGCGGATGCCTTTATGACCAAG R1:TTGGTCATAAAGGCATCCGCAGCTGCGGCGTTGATGTTGG		pmCherry-C1-cRab22a_(M36–41aa)_	pmCherry-C1-cRab22a
42–46	Switch I region	F1: AGGAGCCTCAGCCGCAGCTGCGGCCGTACAGTATCAAAAT R1: GATACTGTACGGCCGCAGCTGCGGCTGAGGCTCCTATTAT		pmCherry-C1-cRab22a_(M42–46aa)_	pmCherry-C1-cRab22a
70–75	β-folds	F1: GGTTTCGTGCTGCCGATGCAGCTGCGGCCAGAGGGTCAGC R1: CTGACCCTCTGGCCGCAGCTGCATCGGCAGCACGAAACCG		pmCherry-C1-cRab22a_(M70–75aa)_	pmCherry-C1-cRab22a
74–75	Switch II region	F1: CTTTAGCTCCAATGGCGGCCAGAGGGTCAGCAGC R1: CTGACCCTCTGGCCGCCATTGGAGCTAAAGCACG		pmCherry-C1-cRab22a_(M74–75aa)_	pmCherry-C1-cRab22a
75	Switch II region	F1: GCTCCAATGTACGCCAGAGGGTCAGCAG R1: GCTGACCCTCTGGCGTACATTGGAGCTA		pmCherry-C1-cRab22a_(Y75A)_	pmCherry-C1-cRab22a
cRab22a ORF		F: GGTCGACCGAGATCTCTCGAGGTATGGCTCTGAGGGAGCTGAA R: ATCCCCGCGGCCGCGGTACGGATCCTCAACAGCAGCTGCGCTTT	Xho I Bam H I	pCMV-Myc-cRab22a	pmCherry-C1-cRab22a

Note: Underlining indicates restriction and mutation site.

## Data Availability

All materials, data, and associated protocols will be made available promptly to readers, without undue qualification, in material transfer agreements.

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
