# Peer review of "Localization of Chicken Rab22a in Cells and Its Relationship to BF or Ii Molecules and Genes"

_animals, 2023, doi:10.3390/ani13030387_

Round 1
Reviewer 1 Report
Yu et al. present a manuscript in which authors investigated the structure of the GTPase RAb22a originated from chicken and its interactions with BF and li molecules which a part of avian antigen presentation.
The manuscript has a very good and detailed introduction that provides a lot of information for readers that are not familiar with Rab proteins. Generally, the manuscript has good approaches. However, after reading, it is not clear what the main aim of this study was as the authors conducted several experiments, but did not explain their purpose and they did not discuss their results very intensively. Also, the experiments do not support the conclusion the authors are drawing. The main aim was described as "investigation of that MHC and li molecules are the key factors that are involved in the activation of antigen presentation". However, none of the experiments actually investigated antigen presentation. Therefore, I have doubts about publishing and suggest intense revision.
I have a few major concerns and several minor comments.
Major
· The title of the manuscript does not fit its content: the colocalization of Rab22a with li was shown in HEK cells and not avian macrophages as the title said; the colocalization to BF was not shown at all; genes do usually not colocalize, but proteins; the manuscript has no data that underlines that HD11 cells are good antigen-presenting cells
· It seems that there is quite some data available for human and murine Rab22a and less for avian Rab22a. However, during reading, it is not clear what was known before and what is actually new data. It feels like almost everything was known before, but had not been shown for avian Rab22a so far. If so, it should be clearly mentioned.
· Generally, conclusions to which no clear evidence is available should be drawn more carefully.
· Further investigations on how cRab22a interacts with li (if they do not co-precipitate) and why downregulation of cRAB22a expression also downregulates expression of BF and li would increase the overall merit of this paper.
Minor
· Lines 18-20 and lines 20-24 have no connection together and do not summarize the manuscript. The authors did not conduct any experiment to examine antigen presentation nor immune cell proliferation.
· Lines 26-27: The same comment as above (regarding the title) applies here.
· Line 34-36: The manuscript gives no evidence for this assumption.
· Line 41: Provide an abbreviation for BF or a short description.
· Line 42: remove “some”
· Line 42-44: This sentence is not inherently consistent.
· Line 44: co-transfection with what? Authors use the words “co-localization” and “co-transfection” but never state which proteins colocalize or which plasmids were co-transfected.
· The abstract should be a short summary of the results. It is not necessary to summarize the methods. A short summary sentence in the end is missing.
· Line 83: pivotal
· Line 122: what are murine RAW 264.7 cells?
· Lines 113-115: Again, authors did not conduct experiments analyzing antigen presentation. The methods and aim of the study are not really clear and does not exactly fit together.
· Line 236: At this point of the manuscript, it is not clear to the reader on which point mutations the authors are referring to.
· Line 237: The structure for the murine rab22a is already available (PDB 1YVD) and should be cited accordingly: Structural basis of family-wide Rab GTPase recognition by rabenosyn-5. Eathiraj, S., Pan, X., Ritacco, C., Lambright, D.G. (2005) Nature 436: 415-419.
· The quality of the confocal images is not good enough to really estimate co-localization. The images are too small, clinched and sort of blurry. I recommend to additionally enlarge a part of the image where co-localization takes place.
· Figure 3: Please clarify in the captions of the confocal images at which time point after transfection the cells were fixed and which cell line was used.
· Line 269: The authors write they want to investigate cli molecules in immune cells. However, they used HEK 239T cells for their experiment. It is not clear why.
· Line 273-275: Please explain where those mutations are coming from that were inserted into Rab22a. How where those sites chosen? Maybe it would be helpful to include a scheme of the gene and were in the gene (which domains) those mutations can be found.
· Lines 351-359: This belongs to the introduction.
· Lines 363-375: This belongs to the introduction.
· Line 382: What is meant by “cRab22a interruption” and how does your data support this suggestion?
· Lines 382-384: The manuscript provides no evidence for this assumption.
· It is generally not clear why human cells were used and not avian cells.
Author Response
Kindly refer to your email sent on 3rd December 2022 regarding the reviewers’ comments on our manuscript (Manuscript Number: animals-2108210) entitled “The features and relationship of chicken Rab22a with BF/Ii at the level of cells, molecules, and genes.” Based on his/ her kind comments, we have carefully modified the manuscript (the revised text is marked with yellow highlighting). We have revised it as per the suggestions of the respectable reviewer.

Reviewer 2 Report
This paper shows that chicken Rab22a localizes to early and late endosomes, similarly to mouse Rab22a, and that chicken invariant chain also co-localizes with it. The authors identified key residues in chicken Rab22a that are required for its localization, and also showed that over- or under-expression of chicken Rab22a affect MHC class I expression, likely through transcriptional regulation by other molecules that are processed via the endosomal pathway (such as invariant chain). Finally, the authors were unable to demonstrate any direct binding between chicken Rab22a and MHC class I or invariant chain. Overall this is a useful study to extend what is known about a Rab protein involved in mammalian antigen cross-presentation to an agriculturally important poultry species.
The title itself is misleading as written, unfortunately, and needs to be changed. At no point did the authors demonstrate that Rab22a co-localizes with MHC class I (we can infer it is likely to, but they did not actually demonstrate this), and none of the co-localization studies were performed in HD11 cells; only the transcription studies. For that matter, nothing in this paper demonstrates that HD11 cells are “superior” at antigen presentation (more so than any other macrophage cell line) or demonstrates their ability e.g. in an antigen presentation assay. By the same token, the first highlights statement is equally misleading.
Questions: Do the primers chosen to measure MHC class I expression amplify both BF-1 and BF-2, or only BF-2?
It would be helpful to provide a diagram or table summarizing the amino acid changes in the mutants generated, including both the functional regions targeted and the amino acids substituted, rather than only providing the primers used to make them.
The last two paragraphs of discussion about how Rab22a affects MHC class I and invariant chain transcriptional expression in the absence of direct protein interactions would benefit from some literature review of potential signaling pathways involved that may be affected by endosome pathway disruption, e.g. the transcription factors downstream of invariant chain-MIF.
Minor issues (most of these appear to be translation problems and I have only highlighted errors that are confusing; some additional overall English editing would improve general readability):
Line #194 “laser confocal microscope (electron microscope)”: this is likely a translation error but it’s evident that electron microscopy was not performed, only laser confocal microscopy.
Line #236 “red pentagram indicates”: the figure is marked with red diamonds, not pentagrams.
Line #261-262 “These results indicate that cRab22a and mRab22a were not localized in early and late endosomes (Figure 3, C and F)”: as written this is a contradictory statement to the main theme of the paragraph (and Figure 3). I expect what the authors mean here is that RFP did not localize to the early and late endosomes in the absence of cRab22a or mRab22a, but this needs to be stated more clearly.
Line #265 should say “the arrows indicate co-localization of molecules”: confocal does not demonstrate that molecules are “linked” through either binding interactions or covalent bonds. Line #304 has the same problem
Line #289 “observed no orange granules”: it looks to me like there is co-localization in Figure 4C, and there’s even an arrow pointing to the orange granules. I assume the “no” is an error?
The labels Figure 5A and 5B are switched in the text.
Author Response

(The authors gave the same response as above.)

Round 2
Reviewer 1 Report
· The main discrepancies regarding the abstract, summary and introduction were accordingly revised. Other points however, where not changed at all, although authors claim in their cover letter that they did change it (see my comments from the first review round):
· Lines 143-145: The text still says: “The current study was designed to investigate that MHC and Ii molecules are the key factors that are involved in the activation of antigen presentation.” However, the authors did not investigate anything related to antigen presentation. Please revise the aim of the study.
Figure 3 and 4: I suggested to make confocal pictures with higher quality and to enlarge parts of the area where colocalization occurs (or not) as it is a main part of the manuscript. Authors claim to have change the pictures, however they did not. It is not necessary to make new pictures. They just need to be edited better and uploaded in a higher resolution.
· Line 315: The text still says: “To understand the functional and structural properties of cRab22a, its binding with related molecules was studied and its involvement in the transportation of cIi molecule in immune cells. “I already noted that no immune cells were used for this experiment and therefore, it is not correct to write it. However, it is still not changed.
· The parts in the discussion that are very introductory are still there although I suggested to move them to the introduction and the authors wrote in their cover letter that they did moved them. Genereal information should be placed in the introduction.
Additionally, the title needs some English editing: The features and relationship of chicken Rab22a with BF/Ii at the cellular, molecular and genetic level
Author Response
Dear Reviewer 1
Journal of animals MDPI.
Subject: Response to Reviewer 1’ Comments.
Kindly refer to your email sent on 14th January 2023 regarding the reviewers’ comments on our manuscript (Manuscript Number: animals-2108210) entitled “Localization of chicken Rab22a in cells and its relationship to BF or Ii molecules and genes.” Based on his/ her kind comments, we have carefully modified the manuscript (the revised text is marked with yellow highlighting and red color writing). We have revised it according to your valuable suggestions. You have provided us again very useful suggestions that have helped us to improve the quality of our manuscript. We hope that the revised manuscript now will meet the standards for publication.
